# Comparative Analysis Revealed Intrageneric and Intraspecific Genomic Variation in Chloroplast Genomes of *Actinidia* spp. (*Actinidiaceae, Viridiplantae*)

Maria Gladysheva-Azgari [1,2,*,†], Fedor Sharko [2,3,†], Natalia Slobodova [1,2,4], Kristina Petrova [2,5], Eugenia Boulygina [2], Svetlana Tsygankova [1,2] and Irina Mitrofanova [1]

1   N.V. Tsitsin Main Botanical Garden, Russian Academy of Sciences, 127276 Moscow, Russia; irimitrofanova@yandex.ru (I.M.)
2   National Research Center "Kurchatov Institute", 123182 Moscow, Russia
3   Research Center of Biotechnology of the Russian Academy of Sciences, 119071 Moscow, Russia
4   Faculty of Biology and Biotechnology, HSE University, 101000 Moscow, Russia
5   Research Center for Medical Genetics, 115552 Moscow, Russia
*   Correspondence: marglader@gmail.com
†   These authors contributed equally to this work.

**Abstract:** About ten species of the genus *Actinidia* Lindl. are known as cultivated plants—alongside the most known *A. chinensis* var. *deliciosa*, there are many others, including Far-Eastern cold-hardy kiwifruits such as *A. arguta*, *A. kolomikta*, and *A. polygama*. Unlike most plant species, in which the chloroplast genome is maternally inherited, the family *Actinidiaceae* possesses a complex system of plastid inheritance with possible transmission through both maternal and paternal lines. The main aim of this work was the assembly of the plastid genomes of three species of *Actinidia*, their comparison with already-available sequences from databases, and evolutionary analyses. We discovered that the gene composition and gene sequences are conserved; the studied species are either subject to purifying selection or not subject to selection at all (with some exceptions, such as the *ycf2* gene). However, the chloroplast chromosomes of some *Actinidia* species have undergone significant structural rearrangements, leading to the persistence of two main forms, both on an intrageneric and intraspecific level. These results expand our understanding of plastid genomics and genetic diversity within the genus *Actinidia*, providing a basis for future research in molecular marker development, phylogenetic analysis, and population studies.

**Keywords:** cpDNA variations in plants; plastid genomes; kiwifruits; phylogenetics of Ericales; DNA barcoding in plants; *ycf2* gene selection process

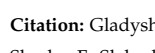



## 1. Introduction

The genus *Actinidia* Lindl., together with the genera *Saurauia* and *Clematoclethra*, belongs to the family *Actinidiaceae* [1,2] and includes more than 60 species [3]. More than ten species of the genus *Actinidia* bear edible fruits with a pleasant taste and high nutritional value; they possess a unique set of biologically active substances and contain a significant amount of ascorbic acid, though they have a low acidity level in terms of acid-sugar balance [4]. The most common and well-known species throughout the world is *Actinidia chinensis* var. *deliciosa* [5]. However, recently, other species of *Actinidia* have become a relatively new type of commercially grown fruit. In most regions of Russia, it is possible to cultivate three Far Eastern species of the genus *Actinidia* [6], perennial cold-hardy vines with smaller (berry-sized) kiwifruits with thin edible skin: *Actinidia arguta* (Siebold ex Zucc.) Planch. ex Miq., *A. kolomikta* (Rupr. ex Maxim.) Maxim., and *A. polygama* (Siebold ex Zucc.) Maxim. Based on the morphological characteristics of species, the genus *Actinidia* was classified into four intrageneric sections: Leiocarpae, Maculatae, Stellatae, and Strigosae [3].

Such classification systems do not always reflect the actual relationships among species [7]; a more accurate approach, which can help solve complex issues, is to take into account molecular data [8,9].

For phylogenetic studies of plants, both individual genes and complete genomes of chloroplasts are often used due to the relatively small plastome size and the presence of conserved regions [10–12]. The *Actinidia* spp. possess a rather conserved genome structure with four independent parts, including an LSC (large single-copy) region, an SSC (small single-copy) region, and two separated inverted repeat regions (IRa and IRb) between the LSC and SSC. According to the NCBI database, chloroplast DNA (cpDNA) among representatives of the genus *Actinidia* is represented by one circular chromosome with a size of about 156 kbp and 130 genes, of which 83 are protein-coding. One of the most notable features of *Actinidia* cpDNA is the loss of part of the *clpP* gene compared to some other members of Ericales [13]. Moreover, despite the high conservation of the plastome in terrestrial plants, sometimes massive structural changes are found in some families of angiosperms, such as *Geraniaceae* [14,15], *Fabaceae* [16–18], and *Campanulaceae* [19,20].

In most plant species, the chloroplast genome is maternally inherited. However, some species are known to be able to transmit their plastid genomes through the paternal line. Such contrasting options of organelle inheritance have been found in some gymnosperms [21], as well as in some angiosperms, for example, *Musa* [22] and *Cucumis* [23]. *Actinidiaceae* show a complex system of plastid inheritance with possible transmission through both maternal and paternal lines [24]. Since some of the restructured angiosperm plastomes occur in lineages with biparental plastid inheritance [25], it can be hypothesized that the nature of inheritance may influence the stability of the chloroplast genome. There are cases in the literature [26] where biparental inheritance in combination with the fusion of paternal and maternal plastids can lead to homologous recombination between different copies of the cpDNA, which ultimately entails a change in the structure of the genome [27]. Restructuring of plastomes is often associated with the presence of small, dispersed repeats (SDRs), although the exact mechanism is still unknown [28–31].

Due to the unusual form of inheritance of chloroplasts and putative heteroplasmy in *Actinidia* species, the main aim of this work was the de novo chloroplast assembly of three *Actinidia* species (*A. arguta*, *A. kolomikta*, and *A. polygama*) from the collection of the N.V. Tsitsin Main Botanical Garden, Russian Academy of Sciences, comparison with already available sequences from databases, and the evolutionary characterization of these species in terms of positive selective processes.

## 2. Materials and Methods

### 2.1. Sampling and Sequencing

Young leaves of 3 *Actinidia* species (*A. arguta*, *A. kolomikta*, and *A. polygama*) were obtained from the collection of Far Eastern Species of the N.V. Tsitsin Main Botanical Garden, Russian Academy of Sciences. Total genomic DNA was extracted according to the method of Lo Piccolo [32] with minor modifications: leaf tissues were disrupted by the TissueLyser II system (Qiagen, Hilden, Germany), centrifugation before isopropanol precipitation was performed at room temperature, and precipitated DNA pellets were dissolved in 30 μL of Milli-Q water. In the case of preparing genomic DNA for sequencing on a GridION device, purification was additionally performed on Genomic Tip 20/G columns (Qiagen, Hilden, Germany) according to the manufacturer's standard protocol. The quality and quantity of DNA were assessed spectrophotometrically in a Nanodrop 1000 device (Thermo Fisher Scientific, Waltham, MA, USA) and using a Qubit fluorometer (Invitrogen, Waltham, MA, USA) using the Qubit ™ dsDNA BR Assay Kit.

To create DNA libraries, the NEBNext® Ultra™ II DNA Library Prep Kit for Illumina® (New England Biolabs, Ipswich, MA, USA) was used according to the manufacturer's protocol. Paired-end sequencing (2 × 150 bp) of the obtained libraries was performed on the high-performance sequencer NovaSeq 6000 (Illumina, San Diego, CA, USA). To carry out the hybrid assembly, sequencing was carried out on a GridION device (Oxford

Nanopore Technologies, Oxford, UK) with a Ligation Sequencing Kit according to the manufacturer's recommendations.

### 2.2. Chloroplast Genome Assembly and Comparative Analyses

Obtained raw reads were trimmed using the fastp program (v0.23.4) [33], and the resulting data were then mapped to the *A. arguta* chloroplast genome (NC_034913.1) from the Refseq database [34] using bowtie2 (version 2.4.4) [35] with the local and minimap2 (v2.24-r1122) options [36] for long reads. The mapped reads were then used for the hybrid assembly of chloroplast genomes. The Unicycler (v0.5.0) program [37] with default parameters was used to generate contigs de novo, order them, and concatenate them into supercontigs. Annotation of the three chloroplast genomes was performed using the PGA program (release: 29 October 2020) [38]. The tRNA genes were predicted with tRNAscan-SE (v2.0.12) [39]. Physical maps were generated using the OrganellarGenomeDRAW (OG-DRAW) tool (v1.3.1) [40], followed by manual modification. The assembled and annotated complete chloroplast genome sequences of the three species of *Actidinia* were uploaded to the NCBI database with the following project number: PRJNA1011506.

For comparative analysis, 19 chloroplast genome sequences of representatives of the genus *Actinidia*, two species of *Clematoclethra scandens* and *Saurauia tristyla* from other genera belonging to the family *Actinidiaceae*, and three species of the genus *Vaccinium* from the order Ericales were added from the GenBank database. The boundary regions of the chloroplast genome in each species were identified and visualized using IRscope (v.0.1) [41], illustrating their contraction or expansion. To assess the conservation of individual genes, we used mVISTA (online version) [42] and visualized the results through two alignment programs: LAGAN [43], which generates multiple alignments regardless of inversions, and Shuffle-LAGAN, which can identify sequence rearrangements and inversions. This allowed for a comprehensive comparison of gene conservation across species.

We also conducted a selective pressure analysis, comparing 22 species of *Actinidia* and 6 species from other genera through pairwise comparisons. To identify positive selection, all protein-coding sequences (CDSs) from each of the 28 species were utilized. Alignment of the CDSs for each gene was performed using mafft (v7.427) [44] for each pair. The Ka/Ks ratio was then estimated separately for each of the 79 genes within the *Actinidia* group. Additionally, a species vs. species Ka/Ks ratio was calculated. The Ka/Ks ratios were determined using the KaKs calculator (v2.0) [45].

### 3. Results

#### 3.1. Organization of the Chloroplast Genomes of de Novo Assembled Actinidia Species

For *A. arguta*, about 39,439,070 raw paired Illumina reads and 2,188,589 Nanopore reads were obtained; for *A. kolomikta*, 68,208,431 paired Illumina reads and 905,244 Nanopore reads; and for *A. polygama*, 71,186,382 and 873,928 Nanopore reads (full statistics for Nanopore reads are provided in Supplementary Table S1). Complete chloroplast genomes of 3 species of the genus *Actinidia* (*A. arguta*, *A. polygama*, and *A. kolomikta*) were assembled de novo. They had a 4-part structure and consisted of large (LSC) and small (SSC) single-copy regions separated by a pair of identical inverted repeats (IR) (Supplementary Table S2). The size of the plastome was ~157,000 bp. 113 genes were annotated, of which 79 were protein-coding genes, 30 tRNA-coding genes (10 of which were represented by two copies), and 4 rRNA genes, each of which was represented by two copies. Maps of the chloroplast genomes are presented in Figure 1 (for *A. arguta*) and Supplementary Figure S1 (for *A. kolomikta* and *A. polygama*).

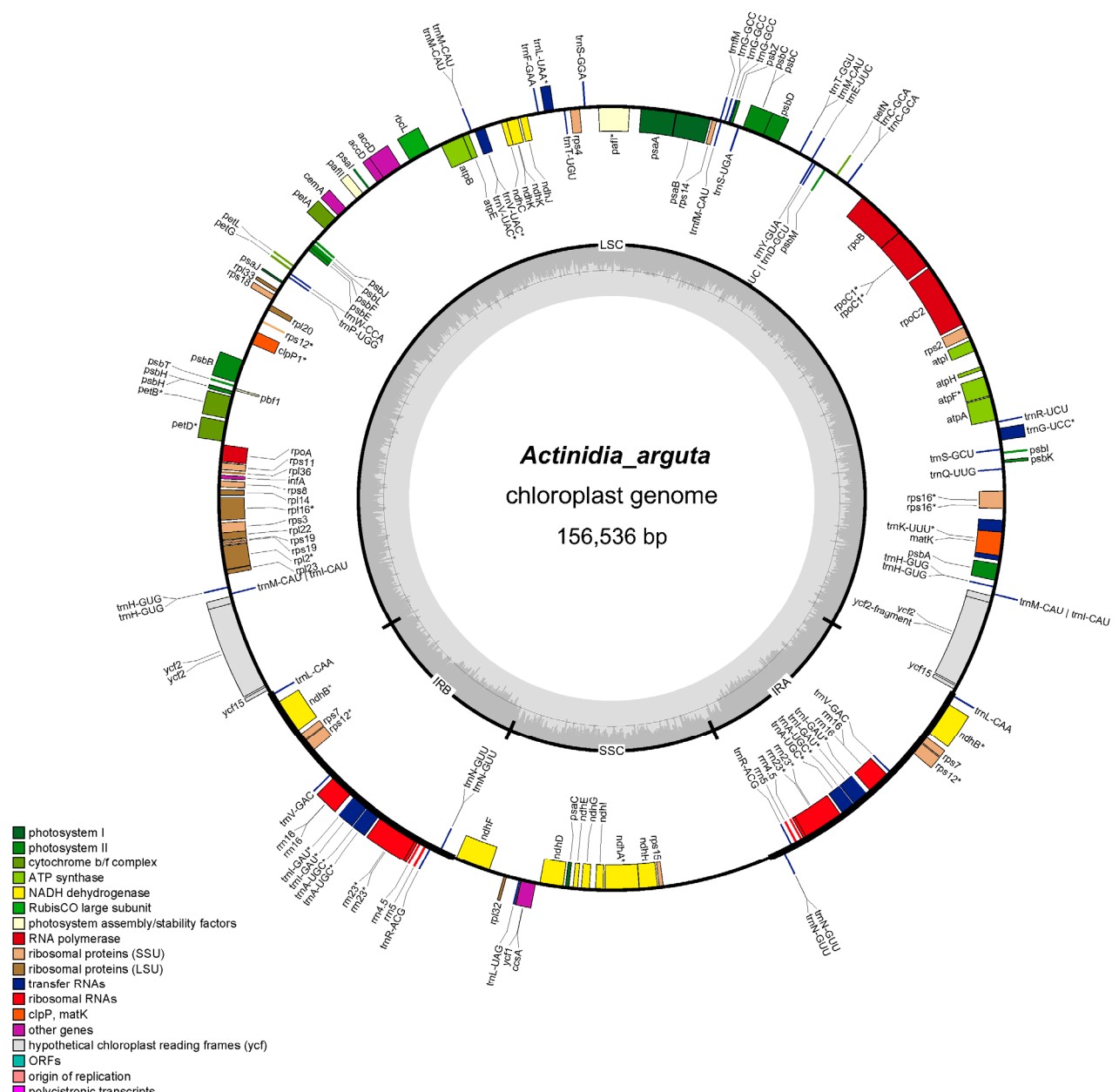

**Figure 1.** Physical map of the chloroplast DNA in *A. arguta*. Genes present in the inner part of the circles are transcribed in a clockwise direction. Genes present in the outer part of the circles are transcribed in a counterclockwise direction. In the inner map, the dark grey region refers to the GC content. Asterisk symbol (*) refers to the genes with introns.

A comparison was also made for gene composition in the studied samples (Supplementary Table S3). As can be seen from Supplementary Table S3, most of the genes are present in the genomes of chloroplasts of all studied species and cultivars (102 genes in sum). The presence of some genes missing from the initial annotation in *A. arguta* (*psbZ*, *pafI*, *rps7*, *rps11*, *accD*, *rps14*, and *ndhD*) was further verified using BLAST. Validation showed that these genes are still present in *A. arguta* samples, but for various reasons they are not annotated automatically; in other cases, genes may not be recognized by open reading frames (ORF).

### 3.2. Boundary Regions and Comparative Analysis

By comparing the chloroplast genomes of *Actinidia* species, the close outgroups *Clematoclethra* and *Saurauia*, and another genus in the order Ericales, *Vaccinium*, we found that gene order near junctions of regions (LSC, SSC, Ira, and IRb) varied greatly, not only between genera, but also within genera and even within species (Supplementary Figure S2). Common characteristic for almost all Ericales species studied are *ndfH* at the IRb/SSC boundary, *trnH* at the LSC/IRb boundary, and *matK* at the IRa/LSC boundary. *Actinidia*, *Clematoclethra*, and *Saurauia* are mostly similar to each other; the main differences are the copy positions of the *ycf2* gene and the sequence of several genes in the LSC region, starting with *rps3* and ending with *rpl23*. For comparison, for the genus *Vaccinium*, *ndhB* and *rps7*, *trnV* on the LSC side, *trnH*, and *rps16* on the IRb side at the LSC/IRb boundary, and *rps16*, *trnH*, and *psbA* at the IRa/LSC boundary.

In the first comparison of boundary regions (Supplementary Figure S2) of the chloroplast genomes of *A. arguta*, *A. kolomikta*, and *A. polygama*, we used only de novo assembled sequences. Separately, we compared the boundary regions for de novo sequences and sequences for the same species from the databases (Figure 2).

## Inverted Repeats

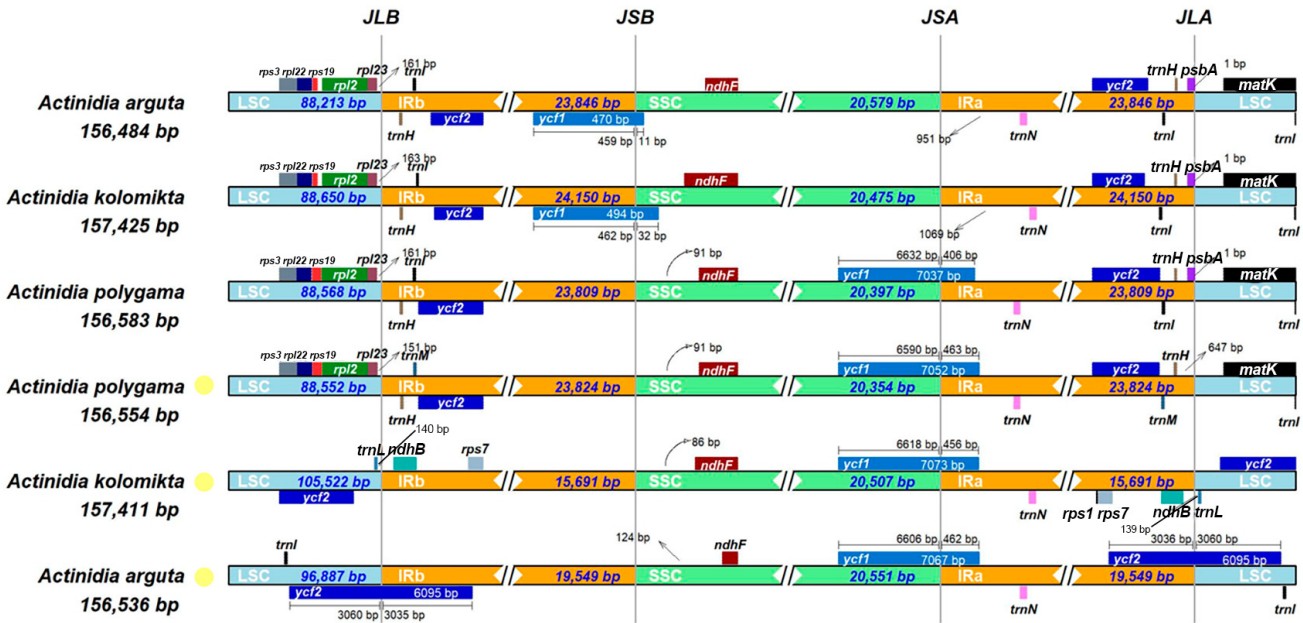

**Figure 2.** Comparison of border regions of the chloroplast genomes of *A. arguta*, *A. kolomikta*, and *A. polygama* assembled de novo and the same species from databases. De novo assembled sequences are marked by yellow dots. Genes are shown as boxes, and the gap between the genes and the junctions of cpDNA regions is indicated by the number of bases unless the gene overlaps with the boundary. Gene extensions are also indicated above the boxes.

The comparison shows that even within the same species, gene order can vary—the sequences from the databases and the de novo assembled *A. polygama* are similar to most other *Actinidia* species, as well as *Clematoclethra* and *Saurauia*, while the de novo assembled sequences of *A. arguta* and *A. kolomikta* are similar to *A. zhejiangensis*.

LAGAN (Supplementary Figure S3) and Shuffle-LAGAN (Supplementary Figure S4) show that plastid genomes within the genus *Actinidia* may differ more from each other than from the outgroups *Clematoclethra* and *Saurauia*. The highest levels of divergence were found for noncoding regions throughout the sequence, with the most variable being the noncoding regions between *rbcL–accD*, *rpl20–psbB*, *ndhF–rpl32*, and *rps15–trn-GUU*.

The coding regions are largely conserved, with some point variations; the most conserved sequences are tRNAs. High variability was noted for the *petD*, *rpl16*, *ycf2*, *ndhA*, and *rpoA* genes. A comparative analysis of the chloroplast genome of some *Actinidia* species shows that its gene composition is highly conserved. Variations are mainly concentrated in non-coding regions, while coding regions show higher conservation.

### 3.3. Selective Pressure Analysis

We calculated Ka/Ks ratios, that is, the ratio of nonsynonymous substitutions (Ka) to the rate of synonymous substitutions (Ks) at the species level, by combining all protein-coding genes into a supermatrix (Supplementary Table S4). In *Actinidia* species and other Ericales studied, the absolute Ka/Ks ratio was approximately $\leq 1$ in most cases (range 0.41 to 0.9). This result indicates that, at the level of protein-coding sequences of the entire chloroplast, the studied species are either subject to purifying selection or not subject to selection at all. The exceptions are comparisons of *A. chinensis*–*A. lijiangensis* (Ka/Ks = 2.2), *A. chinensis*–*A. deliciosa* (Ka/Ks = 2), and *A. valvata*–*A. polygama* (de novo assembly) (Ka/Ks = 1.6). However, in these cases, the *p*-value exceeded 0.05.

Ka/Ks ratios were also calculated for some individual protein-coding genes (Supplementary Table S5). In most cases, the absolute value of Ka/Ks was <1, which may indicate purifying selection or the absence of selection in general if the value was equal to or close to one (concise results are provided in Table 1).

**Table 1.** Ka/Ks ratio for chloroplast genes.

| Gene | Ka/Ks Ratio |
|------|-------------|
| *atpA* | <1, with some exceptions (e.g., for some *Clematoclethra scandens* comparisons) |
| *atpB* | <1 |
| *atpE* | <1 |
| *atpF* | <1, with some exceptions (e.g., for *Actinidia tetramera* comparisons) |
| *atpH* | <1 |
| *atpI* | <1 |
| *ccsA* | <1, with some exceptions (e.g., for some *Clematoclethra scandens* comparisons) |
| *matK* | <1, with some exceptions |
| *ndhA* | <1 |
| *ndhB* | <1 |
| *ndhC* | <1 |
| *ndhD* | <1 |
| *ndhE* | <1 |
| *ndhF* | <1 |
| *ndhG* | <1 |
| *ndhH* | <1 |
| *ndhI* | <1 |
| *ndhJ* | <1 |
| *ndhK* | <1 |
| *petA* | <1 |
| *petB* | <1 |
| *petD* | <1 |
| *petL* | <1 |
| *psaA* | <1 |
| *psaI* | <1 |
| *psbB* | <1 |
| *psbC* | <1 |
| *psbD* | <1 |
| *psbE* | <1 |
| *psbH* | <1 |

**Table 1.** *Cont.*

| Gene | Ka/Ks Ratio |
|------|-------------|
| *psbJ* | <1, with some exceptions (e.g., for some *Saurauia tristyla* comparisons with de novo assembled *Actinidia* cp genomes) |
| *psbL* | <1 |
| *psbM* | <1 |
| *psbT* | <1 |
| *psbZ* | <1 |
| *rbcL* | <1 |
| *rpl2* | <1 |
| *rpl14* | <1 |
| *rpl16* | <1 |
| *rpl32* | <1 |
| *rpl33* | <1 |
| *rpl36* | <1 |
| *rpoA* | <1, with some exceptions (e.g., for some de novo assembled *Actinidia arguta* comparisons) |
| *rpoB* | <1 |
| *rpoC2* | <1, with some exceptions (e.g., for some *Actinidia rufa* comparisons) |
| *rps2* | <1, with some exceptions (e.g., for some *Actinidia polygama* and de novo assembled *Actinidia kolomikta* comparisons) |
| *rps3* | <1, with some exceptions (e.g., for some de novo assembled *Actinidia kolomikta* comparisons) |
| *rps4* | <1, with some exceptions (e.g., for some *Actinidia rubus* and *Actinidia callosa* var. *henryi* comparisons) |
| *rps7* | <1, with some exceptions |
| *rps8* | <1 |
| *rps11* | <1 |
| *rps12* | <1 |
| *rps14* | <1 |
| *rps15* | <1 |
| *rps18* | <1, with some exceptions (e.g., for some *Vaccinium corymbosum* comparisons) |
| *rps19* | <1, with some exceptions |
| *ycf2* | >1, with some exceptions |
| *ycf1* | <1, with some exceptions |

As indicated in Table 1, the *ycf2* gene had a Ka/Ks value > 1 (with exceptions), which suggests possible positive selection. Nevertheless, in the majority of cases with Ka/Ks < 1, the values also fluctuated around 1, indicating the weakness of selective pressure or even the absence of it. A high Ka/Ks value (>3) was observed for a low number of comparisons, which included the *rps4* and *ycf2* genes.

## 4. Discussion

In this study, we sequenced and annotated the cp genomes of three *Actinidia* species and compared the genomic features of *Actinidia* cpDNA. A comparison of the cp genomes of samples from the collection of the Main Botanical Garden with sequences of *Actinidia*, *Clematoclethra*, *Saurauia*, and *Vaccinium* species from the NCBI database revealed not only differences within the order and genus, but also within species.

The de novo assembled cp genomes of *A. arguta*, *A. kolomikta*, and *A. polygama* have a typical circular structure maintaining LSC, SSC, and two IR regions; when compared with genomes from databases, they are conserved in terms of genome length and gene composition. Low Ka/Ks ratios at the chloroplast genome level in *Actinidia* species indicate that most genes have either been subjected to purifying selection to maintain conserved functions or are not under selective pressure at all.

Most protein-coding genes are also conserved, with a few exceptions. Among these exceptions is the *ycf2* gene, which not only exhibits high variability, but also undergoes

positive selection (driving change) and changes its position and proximity to the boundary regions on the plastid genome, which is unusual for usually conserved chloroplast genes. The *ycf1* and *ycf2* genes are the largest chloroplast genes in angiosperms and are essential for the functioning of the plant cell [46]. The presumed function of *ycf2* is to provide ATP for the transfer of nuclear preproteins into the chloroplast, and the function of *ycf1* is related to the TIC complex that facilitates the transfer of proteins in and out through the chloroplast membrane [47]. Ericales is not the only taxon in which *ycf2* undergoes a positive selection process [48–51] and even at the species level [52]. However, the reason for this and the benefit of this strategy compared to gene transfer to the nucleus remain unclear.

Major differences among the chloroplast genomes we obtained affect intergenic non-coding regions, gene order, and their position relative to the boundaries of LSC, SSC, and IRs. Structural rearrangements in chloroplast genomes occur at different taxon levels, for example, at the genus level [53]. It was previously shown that at the species level, rearrangements do not appear to such a strong degree [54,55]—the main layout remains the same, and only the proximity/distance to the boundary regions changes. Rearrangements are in many cases associated with small, dispersed repeats (SDRs), which are thought to contribute to the double-strand break-induced repair mechanism. SDRs often contribute significantly to repeat space in genomes with highly rearranged gene order and add to structural polymorphism even in closely related lineages [26].

Our study shows that large differences in gene order can exist within the same angiosperm species: in our de novo samples of *A. arguta* and *A. kolomikta*, the overall gene order differs from that in the sequences of the same species from the NCBI databases. The general layout of genes near boundary regions of de novo assembled *A. arguta* and *A. kolomikta* is similar not to *A. arguta* and *A. kolomikta* from NCBI, but to *A. zhejiangensis*. However, de novo assembled *A. polygama* is similar to *A. polygama* from NCBI.

There are certain patterns in the arrangement of genes relative to the boundaries of regions in the family *Actinidiaceae* (*Actinidia*, *Saurauia*, and *Clematoclethra*) that support a division of cp genomes into two groups (Figure 3, images near the round bracket).

Between the genus *Vaccinium* and the family *Actinidiaceae*, there is an "intermediate" form—the cp genome of *A. lijiangensis*. Other studied sequences can be divided into two main types according to the position of genes near the LSC/IRb boundary.

The discovery of two main types of cp genomes among the genus *Actinidia*, together with the presence of these two types in representatives of the same species, suggests that at the level of the genus or even the family *Actinidiaceae*, the presence of several types of cp genomes is consolidated, which can be rearranged further by SDR. Presumable mechanisms for the fixation of several types of cp genomes may be the heteroplasmy of *Actinidia* (due to the supposed mixed inheritance of chloroplasts) and the rapid elimination of one of the types during selection or even at the stage of development of the organism. The identification of potential intermediate forms of *Actinidia* with heteroplasmy, the assembly and annotation of cp genomes of other genera of the family *Actinidiaceae*, and the mechanisms of maintaining the presence of two types of cp genomes require further study.

Such large structural changes can greatly complicate phylogenetic analysis, and additional barcode or sequencing studies are required to confirm the exact cpDNA sequence. This study shows that in the case of the genus *Actinidia*, assembly and annotation of the cp genome of a single plant may not be sufficient to characterize the cp genome of an entire species—multiple biological replicates from different spatial ranges must be examined to obtain complete genomic data about species.

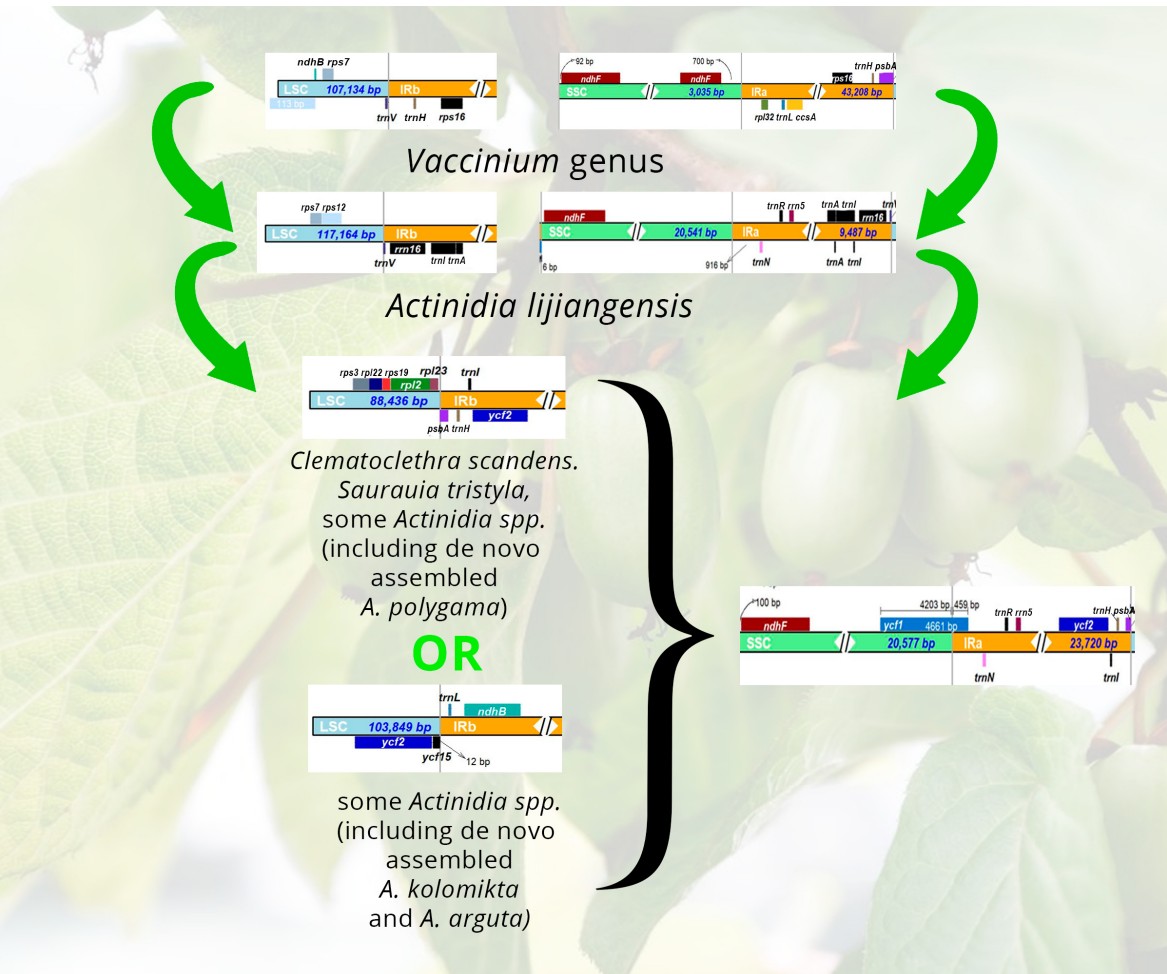

**Figure 3.** Scheme showing variations in the location of genes at the LSC/IRb and SSC/IRb boundaries of cpDNA in studied sequences.

## 5. Conclusions

In this study, we provide an assembly and annotation of the cp genomes of three Far Eastern *Actinidia* species (*A. arguta*, *A. kolomikta*, and *A. polygama*) and also compare the cp genomes of different *Actinidia* species and some other representatives of the order Ericales. The assembled genomes have a typical quadripartite structure and are conserved in the number of genes, but not in their order—we show that in angiosperms, the order of genes in the cp chromosome can differ even at the intraspecific level. In addition, we identified a highly variable protein-coding gene (*ycf2*) that undergoes a positive selection process in the order Ericales. These results expand our understanding of cp genomics and genetic diversity within the genus *Actinidia*, providing a basis for future research in molecular marker development, phylogenetic analysis, and population studies.

**Supplementary Materials:** The following supporting information can be downloaded at: https://www.mdpi.com/article/10.3390/horticulturae9111175/s1, Figure S1: Physical maps of cpDNA of *A. kolomikta* and *A. polygama*; Figure S2: Boundary regions comparison for Ericales; Figure S3: LAGAN analysis; Figure S4: Shuffle-LAGAN analysis; Table S1: Statistics of Nanopore sequencing; Table S2: Overall statistics of the de novo assembly and comparison with sequences from databases; Table S3: Gene composition in de novo assembled sequences; Table S4: Ka/Ks ratios for whole sequences; Table S5: Ka/Ks ratios for each individual gene.

**Author Contributions:** Conceptualization, M.G.-A. and I.M.; methodology, N.S., K.P. and E.B.; software, F.S.; validation, K.P., N.S. and F.S.; formal analysis, F.S.; resources, I.M.; data curation, F.S.; writing—original draft preparation, M.G.-A., N.S. and S.T.; writing—review and editing, M.G.-A., E.B. and F.S.; visualization, F.S. and S.T.; supervision, I.M. All authors have read and agreed to the published version of the manuscript.

**Funding:** This research was funded by the Russian Science Foundation, grant number 22-16-00074.

**Data Availability Statement:** The assembled and annotated complete chloroplast genome sequences of three species of the genus *Actinidia* were deposited in the NCBI SRA database under Bioproject ID PRJNA1011506.

**Acknowledgments:** This work was carried out using high-performance computing resources of the federal center for collective usage at the NRC "Kurchatov Institute", http://computing.kiae.ru/ (accessed on 26 August 2023).

**Conflicts of Interest:** The authors declare no conflict of interest.

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
