# Peer review of "Comparative Analysis Revealed Intrageneric and Intraspecific Genomic Variation in Chloroplast Genomes of Actinidia spp. (Actinidiaceae, Viridiplantae)"

_horticulturae, doi:10.3390/horticulturae9111175_

Round 1
Reviewer 1 Report
Comments and Suggestions for Authors
The manuscript entitled “ Comparative analysis revealed intrageneric and intraspecific genomic variations in Chloroplast genomes of Actinidia spp.” aimed to analyze the genomic variance among chloroplast genomes of Actinidia spp. In this study, the authors assembled de novo the sequence of Actinidia species (A. arguta, A. polygama, A. kolomikta) and then conducted comparative analysis. They found that most gene sequences are conserved. However, chloroplast chromosomes in some Actinidia species have huge structural rearrangements at intrageneric and intraspecific levels. This study provides insights into the understanding of phylogenetic and population studies in the genus Actinidia.
The raw data has been deposited to the NCBI database. The method used in the study is thorough. Conclusions are appropriate, and supported by the data. There are some minor concerns below:
Line 80: delete redundant text “ the collection of ”
Line 82: what kinds of minor modifications that the author did in the methods. Please describe in details.
Line 212: it should be “ Ka/Ks < 1”
Line 214: The author mentioned “the rps gene” has a high Ka/Ks value (>3). It is recommended to add the information to Table 2 or any other supplementary table.
Line 254: what is the function of gene ycf1? Is it similar to ycf2? How about its Ka/Ks value?
Author Response
Thank you for reviewing our paper! Comments are below:
Line 80: changed
Line 82: added: "Total genomic DNA was extracted according to the method of Lo Piccolo [32] with minor modifications: leaf tissues were disrupted by TissueLyser II system (Qiagen, Germany), centrifugation before isopropanol precipitation was performed at room temperature, precipitated DNA pellets were dissolved in 30 μl of Milli-Q water."
Line 212: changed
Line 214: added to Table and changed in text: rps4 gene -> <1, with some exceptions (e.g., for some Actinidia rubus and Actinidia callosa var. henryi comparisons)
Line 254: added to Supplementary Table and changed in text: The presumed function of ycf2 is to provide ATP for the transfer of nuclear preproteins into the chloroplast, the function of ycf1 is related to the TIC complex that facilitate the transfer of proteins in and out through the chloroplast's membrane [49].

Reviewer 2 Report
Comments and Suggestions for Authors
The article reports sequencing, annotation and comparative analysis of chloroplast genomes in three species of the genus Actinidia.
The manuscript reads well, its language is clear and comprehensible - although, a final styling and checkup for grammar- and spelling errors would be desirable).
Some remarks and suggestions for improving the content:
Line 6: (and any further appearances):
"Main Botanical Garden named after N.V. Tsitsin of the Russian Academy of Sciences"
Suggestion to change to: "N.V. Tsitsin Central Botanical Garden, Russian Academy of Sciences"
Lines 30-50 (Introduction):
The genus Actinidia doesn't count as a generally known taxonomy group of plants. (Even though everybody knows kiwifruits.) In favor of the wider audience, the trivial names of the subject species should be mentioned at least at their first appearance in the text. The lesser known species of the genus, A. arguta, A. kolomikta and A. polygama would probably also deserve a very short description concerning their typical morphology features and (potential) commercial values.
Lines 90-93 (Materials and Methods):
Specify the applied sequencing mode (like: "100bp paired-end", etc.) of Illumina short-read sequencing.
Further: Take care of more clear separation of Methods content and Results content. For example, data concerning number of available reads (Illumina, Nanopore) rather belong to the Results section. Also provide some basic metrics (average and median read length) of Nanopore reads in each sample.
Line 97:
I assume you mapped only the short reads onto a reference chloroplast genome, before performing the hybrid assembly. Please, clarify this.
Pages 4-6 (Figures 1-3):
The circular maps show that the chloroplast genomes of the three subject species are highly similar. Therefore, showing all maps in the main text is unnecessary - the second and the third maps could be put into the Supplementary. However, replacing the circular maps with two linear comparative maps would provide visually perceptible information about size- and structural differences between all investigated chloroplast genomes. (When comparing two subject chloroplast genomes respectively to the first genome (A. arguta) as common reference.) Such comparative maps can be produced by tools like the Artemis Comparison Tool (http://sanger-pathogens.github.io/Artemis/ACT/) or the R package genoPlotR (https://genoplotr.r-forge.r-project.org/). Another elegant visualizing alternative could be comparative circular maps generated by the Circos tool (http://circos.ca).
Pages 8-9:
I suggest to move Table 2 to the Supplementary.
Lines 298-299:
"Such large structural changes can greatly complicate phylogenetic analysis and require additional barcode studies."
Unclear statement that needs clarification.
The manuscript reads well, its language is clear and comprehensible - although, a final styling and checkup for grammar- and spelling errors would be desirable).
Author Response
Thank you for reviewing our paper! Comments are below:
Line 6: changed to “N.V. Tsitsin Main Botanical Garden, Russian Academy of Sciences”
Lines 30-50 (Introduction): studied Actinidia species have various trivial names - e.g., both A. arguta and A. kolomikta can be referred as "hardy kiwi" or just as "arguta" and "kolomikta" respectively. Only scientific names are given to avoid potential confusion. Added to Abstract: “there are many others, including Far-Eastern cold-hardy kiwifruits such as A. arguta, A. kolomikta and A. polygama.”; added to Introduction: “In most regions of Russia, it is possible to cultivate three Far Eastern species of the genus Actinidia [6], perennial cold-hardy vines with smaller (berry-sized) kiwifruits with thin edible skin: Actinidia arguta (Siebold ex Zucc.) Planch. ex Miq., A. kolomikta (Rupr. ex Maxim.) Maxim. and A. polygama (Siebold ex Zucc.) Maxim.”
Lines 90-93 (Materials and Methods): added: “Paired-end sequencing (2 × 150 bp) of the obtained libraries was performed on the high-performance sequencer NovaSeq 6000 (Illumina, USA).” The paragraph with read number moved to Results. Added: New Supplementary Table S1 with Nanopore statistics.
Line 97: added: “The Unicycler program [37] with default parameters was used to assemble contigs generate contigs de novo, order them and concatenate into supercontigs.”
Pages 4-6 (Figures 1-3): Figure 2 and 3 removed from the main text; new Supplementary Figure S1 contains two comparisons: A. arguta vs A. kolomikta and A. arguta vs. A. polygama.
Pages 8-9: We decided to leave Table in the main text – it displays the result of Ka/Ks ratio calculations and it’s quite concise comparing to tables in Supplementary Materials.
Lines 298-299: added: “Such large structural changes can greatly complicate phylogenetic analysis and require additional barcode or sequencing studies to confirm the exact cpDNA sequence.”

Reviewer 3 Report
Comments and Suggestions for Authors
I read Gladysheva-Azgari et al., ”Comparative analysis…” with interest. It is clearly an interesting topic.
Line 3. “variations” > “variation”
3, 31. “Actinidia spp.” > “Actinidia spp. (Actinidiaceae, Viridiplantae)” – not all readers will know this genus.
14. I propose “the genus Actinidia”
16. “like” > “such as”
17. I propose “The family Actinidiaceae possesses”
19. I propose “assembly of the plastid genome of three species of Actinidia”
20. The authors sometimes use the Oxford comma, sometimes not. Please resolve this in a consistent way throughout the manuscript. If indeed the Oxford comma is to be used, then “databases and” should be “databases, and” on this line.
22. “such as” > “such as the”
22. “However,” > “However, the”
23. “underwent huge” > “has undergone significant”
35. I propose “though being of low acidity level”. But does “low acidity” mean “low pH” = acid or “high pH” = around 7 ?
38. Should “cultivate” be “cultivate two”? If not, then please clarify this sentence.
42. I propose “the genus Actinidia”
42. “sections, “ > “sections: “
43. I propose “Such classification systems do not always”
51. Please cite BCNI formally: https://academic.oup.com/nar/article/51/D1/D29/6825348
53. “83 genes” > “83”
57. I propose “, and Campanulaceae”.
59. “through paternal” > “through the paternal”
72. “de novo assembly” > “de novo chloroplast assembly”
74. I propose “Main Botanical Garden, comparison”
75. I propose “, and the evolutionary “
75. “species-possible” > “species in terms of”
76. The phylogenetic aspect of the study is very poor, like I will get back to later on. I’d say it should be deleted from the study.
79. I propose “, and A. polygama”
79. When the authors say “A. polygama”, do they mean https://www.ipni.org/n/60472666-2 or https://www.ipni.org/n/828406-1? Bottom line, please add author names upon first mention of these three species.
80. I propose “Main Botanical Garden, Russian Academy of Sciences”
86. “by” > “in a”
90. “on a” > “on the”
97. I propose “, and for A. polygama 71,186,283 Illumina and 873,928 Nanopore reads”
98. “A. arguta” should be given in italics.
98. “fastp” – in the interest of scientific reproducibility, all software tools should be specified with version number.
100. “readings” > “reads”? If not, please clarify.
106. I propose “of the three species of Actidinia were uploaded to” – nice to see the data made available like this!
112. “depicted” > “visualized” or “identified and visualized” ?
113. “the contraction” > “their contraction” ?
133. “utilized” > “used”
122. “mafft” (=MAFFT) should be cited formally upon first mention, not at the second mention (line 130).
135. “Chloroplast Genomes” > “chloroplast genomes”
136. I propose “polygama, and”
140. I propose “(10 of which were represented by two copies), and 4 rRNA”
142. I propose “kolomikta, and”
143. “in the” > “in Figures 1-3, respectively”.
144 and on. In the figures, an underscore (“_”) is used inside the species names. Why?
158. Please define what is meant by “boundary region”
161. Please clarify “varied greatly in gene representation”
176-169. Please clarify this sentence. I cannot follow it.
170. “as” > “of”? If not, then what is intended?
177. “marked” > “are marked”
186. “out groups” > “outgroups”
195. I propose “, that is, the ratio”
196. “level by” > “level, by”
210. “in the” > “in”
225. The phylogenetic part of the study has many problems. To begin with, the authors based their taxon sampling on whatever species of Actinidia that have had their chrolorplasts sequenced. This is a violation of sound taxon sampling (https://academic.oup.com/sysbio/article/51/4/588/1698771).
The violate the MIAPA standard ( https://pubmed.ncbi.nlm.nih.gov/16901231/ ) by withholding the multiple sequence alignment + phylogenetic (Newick) tree file. These should be made available for review and to the reader through a TreeBase, Dryad, or supplementary item release. In the interest of scientific reproducibility (https://www.nature.com/articles/d41586-019-00067-3), obviously.
All the usual baseline data is missing. How long was the multiple sequence alignment? How many of the bases were invariable? Variable but parsimony uninformative? Parsimony informative?
In Figs 5-6, all species names should be given in italics. (This goes for the supplementary items, too.)
Why, for some species in Fig 6, is the genus name abbreviated?
The authors do not compare their phylogenetic estimates of Actinidia to that of other phylogenetic studies of the genus (e.g., https://doi.org/10.1043/0363-6445-27.2.408 and DOI 10.17660/ActaHortic.2011.913.6). In fact, the authors don’t use their phylogenetic results for anything. Not a single analysis uses the tree, and not a single conclusion is based on the tree.
Bottom line, I feel that the phylogeny part is compromised, very weak and needless. It should be removed from the study.
232. “species” > “species of”. These are genera, not species.
239. “constructed” > “inferred”
240 “for studied” > “for the studied”
241. “with the” > “with”
255. I propose “The presumed function is ycf2 is to provide”
259. “to nucleus” > “to the nucleus”
259-260. Before a genetic marker is recommended as a DNA barcode, an analysis must be done to see if the genetic marker is flanked by appropriately conserved primer sites. All too many “potential DNA barcodes” have been ruled out due to lack of good primer sites. If the authors want to promote a new DNA barcode, such an analysis of potential primer sites should be done.
271, 308. Delete “for the first time”.
305. I propose “, and A. polygama”
The list of references comes across as very untidy, bordering on the chaotic.
Journal names are sometimes abbreviated (ref 2), sometimes not (ref 6).
For many references, volume/issue/page information is provided (ref 7). For others, it is not (ref 8).
In some reference titles, key nouns and verbs are given with leading uppercase letters (ref 13). In others, not (ref 12).
When journal names are abbreviated, dots are sometimes used (ref 2), sometimes not (ref 14)
Species and genus names are sometimes given in italics in the list of references, sometimes not.
398. “dna” > “DNA”
Supplementary item 1 “Inverted Repeats” > “Inverted repeats”
Comments on the Quality of English Language
See my letter to the authors.
Author Response
Thank you for your thorough review! Comments are below:
Line 3. changed
3, 31. changed
- changed
- changed
- changed
- changed
- changed here and below
- changed
- changed
- changed
- changed and added: “they possess a unique set of biologically active substances and contain a significant amount of ascorbic acid though being of low acidity level in terms of acid-sugar balance [4].”
- changed
- changed
- changed
- changed
- changed
- changed
- changed
- changed
- changed
- changed
- changed
- changed
- deleted
- changed
- Authors’ names are already mentioned above (see lines 39-40)
- changed
- changed
- changed
- changed
- changed
- added: “using the fastp program (v0.23.4)”
- changed
- changed
- changed to “identified and visualized”
- changed
- changed
- the paragraph with the second mention of MAFFT (about the phylogenetic part) is deleted, the reference is moved
- changed
- changed
- changed
- changed
- changed (Figures 2 and 3 became a Supplementary Figure S1 on the advice of another reviewer)
- during bioinformatical data processing the samples were assigned with names with underscores
158 and 161. changed to “By comparing the chloroplast genomes of Actinidia species, the close outgroups Clematoclethra and Saurauia, and another genus in the order Ericales, Vaccinium, we found that gene order near regions (LSC, SSC, IRa and IRb) junctions (boundary regions) varied greatly, not only between genera, but also within genera and even within species (Supplementary Figure S2).”
176-169. changed to “Genes are shown as boxes and the gap between the genes and the junctions of cpDNA regions is indicated by the number of bases unless the gene overlaps with the boundary. Gene extensions are also indicated above the boxes.”
- changed
- changed
- changed
- changed
- changed
- changed
- deleted
- deleted
- deleted
240 changed
- changed
- changed
- changed
259-260. deleted
271, 308. deleted
- changed
References: are brought to a uniform standard where possible – some journals names generally accepted abbreviations.
Supplementary Figure S1: can’t be changed – the program gives the result as an uneditable picture

Round 2
Reviewer 3 Report
Comments and Suggestions for Authors
The revised manuscript is a lot better, and a lot easier to read too.
Line 83. “known” > “well-known”
142. “Actinidia genus” > “the genus Actinidia”
148. “due to” > “due to the”
157. “within the families” > “in some families of” ? Because it is not found in all families, right?
165. “it is possible to speculate” > “it can be hypothesized”
188. “precipitated” > “and precipitated”
257-295. I n my previous review I pointed out to the authors that in the interest of scientific reproducibility, all software tools should be specified with version number. The authors only changed one such case, though (namely on line 257). All other software tools – and there are many – are still not specified with version number. The authors should go through each sentence on 257-295 very carefully: if the sentence mentions a software tool, then a version number should be specified. We’re talking 10+ cases here.
257. “the resulting” > “and the resulting”
262. “concatenate” > “concatenate them”
271. “three more” > “three”
272. “GeneBank” > “GenBank”
379. “regions … junctions” > “junctions of regions (LSC, …)
453. I propose: “. However, in these cases”
559. “the function” > “and the function”
560. “facilitate” > “facilitates”
560. “chloroplast’s” > “chloroplast”
565-566. Please clarify this sentence. What “Major changes” are these? “The major differences among the chloroplast genomes we obtained…”?
569. “only the” > “and only the”
577. “species from” > “species from the”
582. I propose: “) to support a division of cp genomes into two groups”
583. “conditional” should be removed or clarified.
594. I propose: “further by SDRs”.
602. I propose: “, and additional barcode or sequencing studies are required to”
647. “conservative” > “conserved”
652. “among” > “in”
Comments on the Quality of English Language
Dear Sirs/Madams,
The revised manuscript is a lot better – I was happy to see that the authors used the feedback to improve the manuscript. This time I offer only a few minor suggestions.
I give this piece a “Minor revision”.
Disclaimer: I don’t know any of the authors. I am not working on any similar study at the present time. Please feel free to revise my letter to the authors in any way you see fit. I wish to remain anonymous.
Yours sincerely,
The reviewer
Author Response
Thank you for your review! Comments are below:
Line 83. changed
- changed
- changed
- changed
- changed
- changed
257-295. added; LAGAN/Shuffle-LAGAN analyses are the part of mVISTA software
- changed
- changed
- changed
- changed
- changed
- changed
- changed
- changed
- changed
565-566. changed
- changed
- changed
- changed
- deleted
- changed
- changed
- changed
- changed